# The Third Dimension of Eye Care: A Comprehensive Review of 3D Printing in Ophthalmology

Neil Lin [1], Maryse Gagnon [2] and Kevin Y. Wu [3,*]

1   Faculty of Medicine, University of Toronto, Toronto, ON M5S 1A8, Canada; neil.lin@mail.utoronto.ca
2   Faculty of Medicine and Health Sciences, University of Sherbrooke, Sherbrooke, QC J1H 5N4, Canada; maryse.gagnon3@usherbrooke.ca
3   Division of Ophthalmology, Department of Surgery, University of Sherbrooke, Sherbrooke, QC J1G 2E8, Canada
*   Correspondence: yang.wu@usherbrooke.ca

**Abstract:** Three-dimensional (3D) printing is a process in which materials are added together in a layer-by-layer manner to construct customized products. Many different techniques of 3D printing exist, which vary in materials used, cost, advantages, and drawbacks. Medicine is increasingly benefiting from this transformative technology, and the field of ophthalmology is no exception. The possible 3D printing applications in eyecare are vast and have been explored in the literature, such as 3D-printed ocular prosthetics, orbital implants, educational and anatomical models, as well as surgical planning and training. Novel drug-delivery platforms have also emerged because of 3D printing, offering improved treatment modalities for several ocular pathologies. Innovative research in 3D bioprinting of viable tissues, including the cornea, retina, and conjunctiva, is presenting an avenue for regenerative ophthalmic therapies in the future. Although further development in printing capabilities and suitable materials is required, 3D printing represents a powerful tool for enhancing eye health.

**Keywords:** 3D printing; bioprinting; ophthalmology; prosthetics; implants; anatomy; surgery; tissue engineering





## 1. Introduction

Three-dimensional (3D) printing has revolutionized hardware manufacturing as a tool for rapid prototyping and product development. As a form of additive manufacturing, 3D printing fabricates custom constructs layer by layer based on models produced in computer-aided design (CAD) software. In the medical context, images obtained from computed tomography (CT), magnetic resonance imaging (MRI), and 3D scanning may be used to generate patient-specific designs and devices. Key advantages of 3D printers for medical use include the ability to produce complex geometries and the flexibility of using one printer to tailor many diverse designs [1]. These attributes make 3D printing suitable for personalized medicine as printers can cost-effectively produce specialized patient-specific products compared to conventional techniques [2]. Three-dimensional printing has already been established in many medical domains, where it is used to manufacture products from guides for dental surgery to medical instruments to custom orthopedic implants [3,4].

The potential of 3D printing for precision medicine is further illustrated by its current and potential applications in ophthalmology. Notably, the advancement of printing technologies to enable printing on the nanometer-micrometer scale has increased the utility of 3D printing for ophthalmic devices requiring fine detail, such as intraocular implants, ocular prosthetics, and surgical devices. For medical education, 3D-printed anatomical models can express the subtle anatomical details of the eye to better train students and physicians [5]. Similarly, precision surgical treatment, enabled by 3D-printed surgical guides, can help ophthalmologists reduce operative time and enhance patient outcomes [6–8]. Finally,

the emerging field of bioprinting is opening avenues for regenerative medicine in ophthalmology with novel research in manufacturing artificial corneal, retinal, and conjunctival tissue models that could yield sight-restoring treatments in the future [9,10].

This review will first provide a background into 3D printing technologies, followed by connections between these technologies and their exciting applications in ophthalmology.

## 2. Build Instructions for 3D Printing

1. **Production of a computer-aided design (CAD) model:** Each 3D printing project begins with a digital model of the intended product. This model is typically created using computer-aided design (CAD) software or by generating a 3D representation based on CT or MRI scans of an existing object. The geometric data of the 3D model can be stored in a standardized .STL or .OBJ file format (Figure 1).

2. **Slicing and preparation of print file:** Slicing software is then utilized to divide the 3D model into thin horizontal cross-sections, creating a digital representation of each layer to be printed. This step enables the specification of layer height, infill density, and print speed. If the design contains layers that overhang previous layers, it may require the use of support structures, which can be inserted by slicing software to maintain structural integrity against gravity.

3. **Toolpath Generation and Output:** Based on the shape of successive layers, slicing software will generate toolpaths for the 3D printer that contain ordered coordinate directions to create the surface geometry, interior fill pattern, and support structures. Toolpaths are converted into control code specific to the 3D printer that guides the printing process.

4. **Material Selection and Machine Setup:** The printer will be loaded with material and prepared for the physical build process. With the material diversity afforded by 3D printing, the appropriate material should be selected for the desired functional properties of the product.

5. **3D Printing Process:** The printer will then work layer by layer to additively manufacture the product. The 3D printer largely automates this step, and many have integrated control units that monitor the printing process and alert the user if an issue requiring intervention occurs. Once printing is completed, further steps may include product separation from the build platform and safe handling precautions.

6. **Postprocessing:** After printing, the 3D model may require postprocessing, such as polishing, further curing, chemical treatment, coloring, and the removal of support structures, depending on the target function and appearance requirements. Support structures may be removed manually or via dissolution with a targeted solvent.

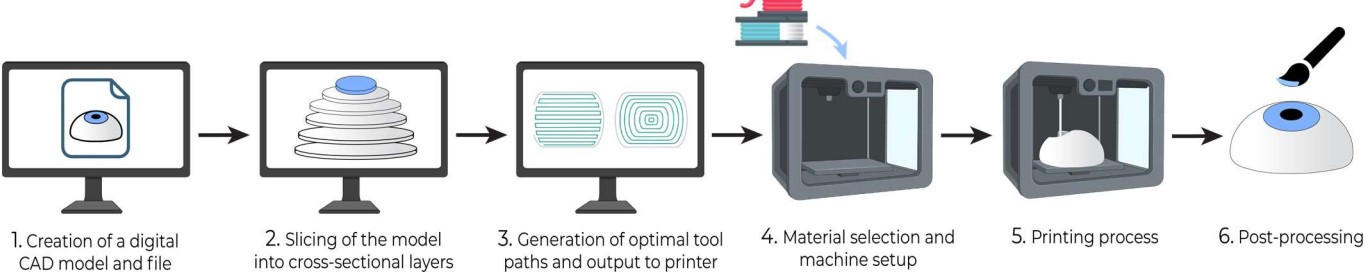

**Figure 1.** General build steps for 3D printing.

## 3. Three-Dimensional Printing Techniques

The techniques of 3D printing are diverse and continually being innovated. Presented here are primary techniques with prevalent uses in industry and medicine (Table 1). A focused summary of key techniques used for ophthalmic applications will be reviewed.

**Table 1.** Summary of 3D printing techniques and their applications in ophthalmology. FDM = Fused Deposition Modeling; SLA = Stereolithography; SLS = Selective Laser Sintering; DMLS = Direct Metal Laser Sintering.

| Printer | FDM | SLS | DMLS | Binder Jetting | SLA | Material Jetting |
|---|---|---|---|---|---|---|
| **Technique** | Extrusion-based | Powder bed fusion | Powder bed fusion | Powder binding | Vat-polymerization | Inkjet droplet deposition |
| **Materials [2,3,11]** | Thermoplastic, composites | Thermoplastic, ceramics, composites | Metal alloys | Thermoplastic, Metals, Ceramics | Photopolymer resin | Photopolymers, waxes |
| **Base Technology [2,3]** | 3D articulating print head depositing heated filament | Laser (e.g., $CO_2$) reflected by mirrors onto powder to sinter it solid | High-power laser (e.g., YAG fiber) that sinters metal powder solid | Print head deposits binder to adhere powder layer by layer | Selective solidification of resin by ultraviolet (UV) source | Deposition of photopolymer droplets that are UV flash-cured |
| **Machine Cost [11]** | Low-Medium | Medium-High | High | Medium | Low-High | Medium-High |
| **Material Cost [11]** | Low-Medium | Low | High | Low-Medium | Medium-High | High |
| **Typical Resolution [3,11]** | 100–150 μm | 50–100 μm | 50–100 μm | 100 μm | 25–75 μm | 25–40 μm |
| **Benefits [2,3,12]** | 1. Good structural strength 2. Inexpensive 3. Capable of multi-material printing 4. Large scalable build volume 5. Widespread and accessible | 1. Good accuracy and detail 2. High strength and durability 3. Suitable for complex parts with internal geometries 4. No support structures required 5. Material variety | 1. High accuracy and precision 2. Suitable for complex parts with internal geometries 3. High strength and durability 4. No support structures required | 1. Cost-effective compared to SLS and DMLS 2. Suitable for complex parts with internal geometries 3. Fast print speed 4. Multiple color printing 5. Material variety | 1. Excellent resolution 2. Best surface finish (smooth) 3. Suitable for complex parts requiring fine detail 4. Print uniformity and isotropy 5. Fast print speed | 1. High repeatability and precision 2. Controllable transparency and color 3. Excellent resolution 4. Capable of multi-material printing |
| **Drawbacks [2,3,12]** | 1. Slow printing time 2. Rough surface finish with anisotropy 3. Requires support structures 4. Lower dimensional accuracy | 1. Expensive 2. Rough surface finish 3. Requires postprocessing to separate part from powder | 1. Very expensive 2. Often requires postprocessing and surface finishing 3. Slow printing time | 1. Inferior strength compared to SLS and DMLS 2. Relatively lower resolution 3. Rough surface finish | 1. Moderate strength 2. Long-term stability reduced by UV sensitivity of resin material 3. Relatively high cost | 1. Relatively weak strength prints 2. Lower temperature resistance 3. Requires support structures and postprocessing |
| **Applications in Ophthalmology [4,13–16]** | Anatomical Models, Prostheses, Surgical Planning Models | Anatomical Models, Prostheses, Surgical Instruments, Implants | Surgical Instruments, Surgical Guides, Implants | Anatomical Models, Surgical Guides and Preplanning Models, Prostheses | Surgical guides, Anophthalmic Socket Conformers, Prostheses | Anatomical Models, Surgical Guides and Preplanning Models, Prostheses |

### 3.1. Extrusion-Based Printing

The most widespread form of extrusion-based printing is Fused Deposition Modeling (FDM), also known as Fused Filament Fabrication. FDM is a mature technology based on the extrusion of thermoplastic or polymer composite materials. Underlying FDM is a mobile print head that heats thermoplastic to a semi-fluid state before depositing it continuously to form the solid shape of a layer once the material cools. This process is repeated layer by layer, with continued deposition and fusion of thermoplastic until the 3D-printed object is complete (Figure 2). Where support structures are needed, an FDM 3D printer will integrate them with each layer in a manner that facilitates later removal. FDM is widely used and accessible due to the low cost of printers and materials, with diverse applications in medicine. Furthermore, it is a flexible technology that can print with multiple materials and colors by changing filament mid-process [17]. Resolution for extrusion-based printing is proportional to nozzle diameter and the movement precision of the nozzle but is relatively low among 3D printing techniques. The melting-based layer adhesion in FDM creates an inherent anisotropy, which is a drawback for strength uniformity and watertightness. In addition, printing time increases with model volume, feature complexity, and infill density [11]. The principles behind extrusion-based printing are also suitable for the deposition of cells and biological scaffolds in the emerging field of bioprinting [18].

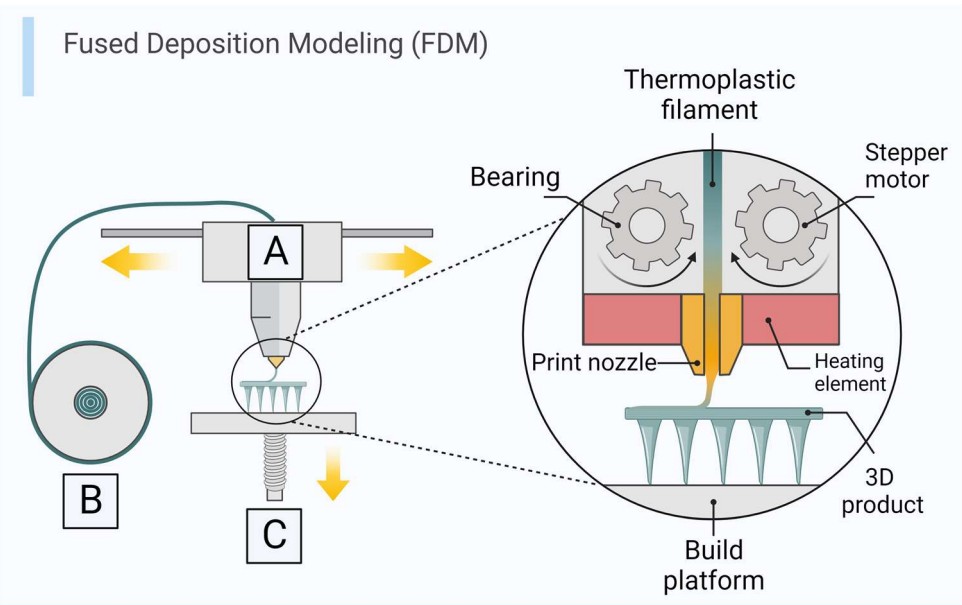

**Figure 2. Fused Deposition Modeling (FDM)** 3D printing with primary components, (A) mobile extrusion print head with integrated heating element, (B) Spool of print material, (C) build platform/print bed.

### 3.2. Powder Bed Fusion Printing

The major forms of powder bed fusion printing are selective laser sintering (SLS) and direct metal laser sintering (DMLS). In these techniques, fine powder particles (<100 μm in diameter) are bound with a precisely controlled laser in specific geometries on one layer of the powder bed [3]. Once the printing of a layer is complete, the bed lowers by the thickness of one layer, and a roller or blade distributes a new flat layer of powder onto the bed surface [11]. The laser sintering is then repeated to build subsequent layers (Figure 3). After being built layer by layer, printed objects are collected from within the powder bed. No support structures are required since the powder surrounding the printed object provides support throughout the printing process. As a result, complex geometry and interior architecture can be created with SLS printing with high strength, uniformity,

and stiffness. DMLS is a specialized form of SLS that utilizes metal powders exclusively with a high-power laser. Powder bed fusion often results in a rough surface, so achieving a smooth finish requires postprocessing, such as media blasting and polishing. Resolution for SLS and DMLS in the X-Y plane is dependent on the laser properties, while Z-resolution is determined by the chosen layer thickness [11]. Powder fusion achieves good resolution, as fine as 50 microns, but build volume is constrained by the size of the powder bed [3]. Larger builds require substantially more powder, which contributes to the expense of powder bed fusion along with the high printer costs. Print speed is fast among printing techniques and can be minimized by reducing the number of layers and orienting the shortest axis of the 3D model in the Z-direction [11].

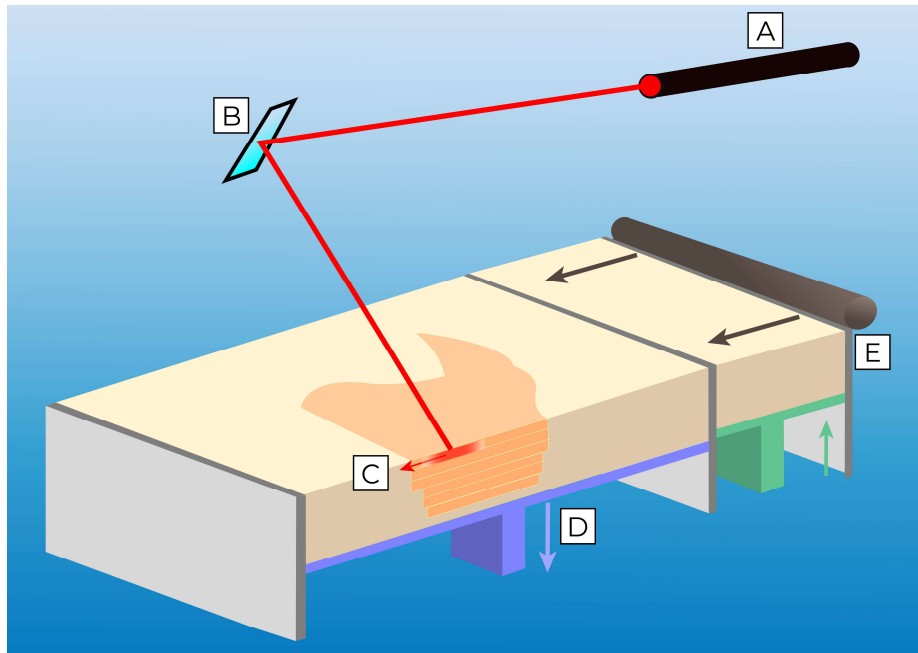

**Figure 3. Selective laser sintering (SLS)** with primary components, (A) laser and optics, (B) X-Y scanning mirror to direct light, (C) powder bed with sintered model embedded and region undergoing fusion in red, (D) build platform that moves vertically for new layers, (E) powder delivery system and roller for new layer distribution.

*3.3. Binder Jetting*

Binder jetting is a 3D printing technology that uses a powder bed similar to powder-sintering methods, but instead of fusing the powder with a laser, binder jetting leverages a liquid binding agent to selectively join powder particles [19]. A highly mobile inkjet head deposits droplets of binding agent to form specific portions of the flat powder bed into layer shapes [14]. The build platform then moves down to allow a roller to deposit another layer of powder before the binding process repeats for the subsequent layer (Figure 4). Binder jetting operates at lower temperatures relative to powder-sintering methods and is comparatively less costly while enabling a large material variety. The lack of rapid heating and cooling during printing generates a uniform grain microstructure that can have superior isotropy compared to power fusion techniques when printing metals. However, the overall functional strength of binder jetting products is inferior to that of SLS and DMLS due to higher porosity [19].

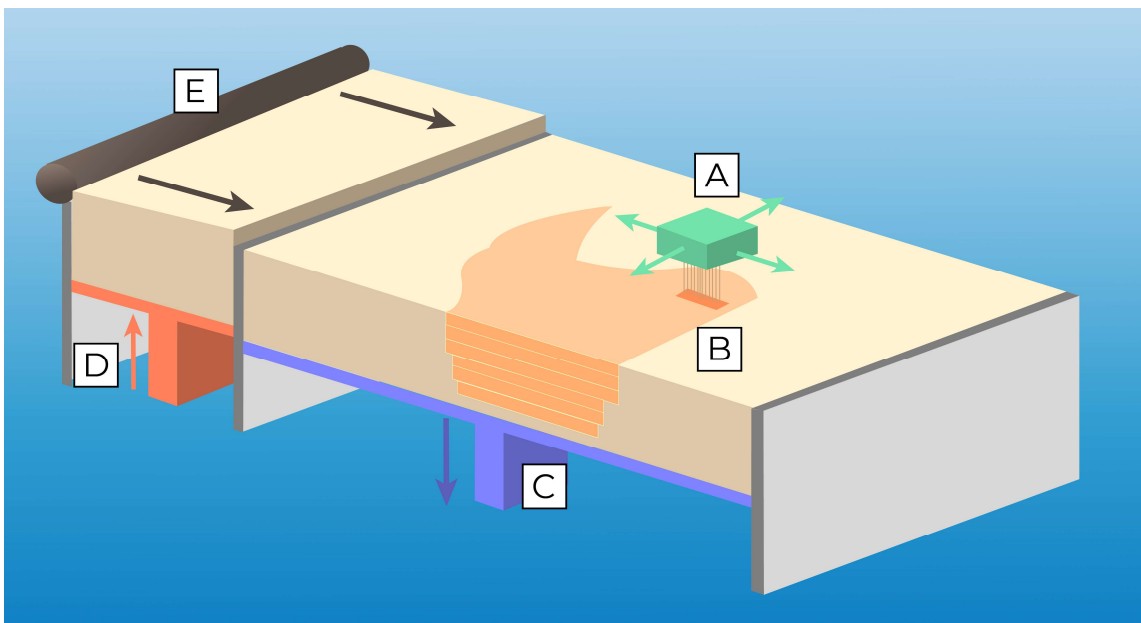

**Figure 4. Binder jetting** 3D printing with primary components, (A) X-Y mobile inkjet head selectively depositing binder adhesive, (B) printed model supported within powder bed, (C) vertically mobile build platform, (D) powder delivery system to supply new layers, (E) roller for powder distribution and leveling.

### 3.4. Vat-Polymerization Printing

Vat-polymerization technology is based on selectively curing a liquid photosensitive resin into a solid using a directed light source. A vat of photosensitive polymer resin is selectively exposed to a precisely controlled laser beam or light projection that outlines the shape of an individual layer. The 3D print is then moved vertically in the vat to allow for the next layer to be polymerized. Stereolithography (SLA) is the most common form of vat-polymerization printing. It utilizes a mirror-directed UV laser to polymerize the shape of specific layers in X and Y dimensions (Figure 5). The printing platform moves in the Z-axis to allow for other layers to be printed [3]. The continuous liquid interface production (CLIP) approach is a subtype of SLA that builds from the top-down with an optical window at the bottom of the resin vat. Diffusion of oxygen through the window prevents adherence to photopolymer as the model is continuously pulled upward from the resin with each layer [20]. Once printing is complete, the liquid resin is drained, and the product is separated from the printing platform. CLIP achieves rapid printing speeds and is more cost-effective than traditional SLA, which requires a large resin vat into which to lower the model. The curing of liquid resin using a precisely controlled light source enables SLA to achieve excellent printing resolution and smooth surface finishes. The chemical crosslinking of layers with covalent bonds also produces isotropy and uniform strength in all directions. However, SLA is limited by the lower mechanical strength of prints and necessary support structures, which must be removed during postprocessing [3]. Other forms of vat-polymerization include digital light processing (DLP), which uses a projector light source to cure entire layers at once. DLP achieves among the fastest 3D printing speeds but is limited in print size and resolution by its light source [21].

### 3.5. Material Jetting

Material jetting is a 3D printing technique that utilizes a specialized inkjet head with multiple nozzles to elute droplets of photopolymer. The droplets are deposited precisely before UV light selectively flash-cures them into the shape of a solid layer. The build platform moves downward to accommodate the next layer's thickness, and the process is repeated (Figure 6). Material jetting is capable of high-resolution isotropic printing with the

expression of fine details while leaving a smooth surface finish [11]. Two primary material jetting processes are multi-jet modeling (MJM) and PolyJet, which are fundamentally similar to differences in material choice and postprocessing methods [22].

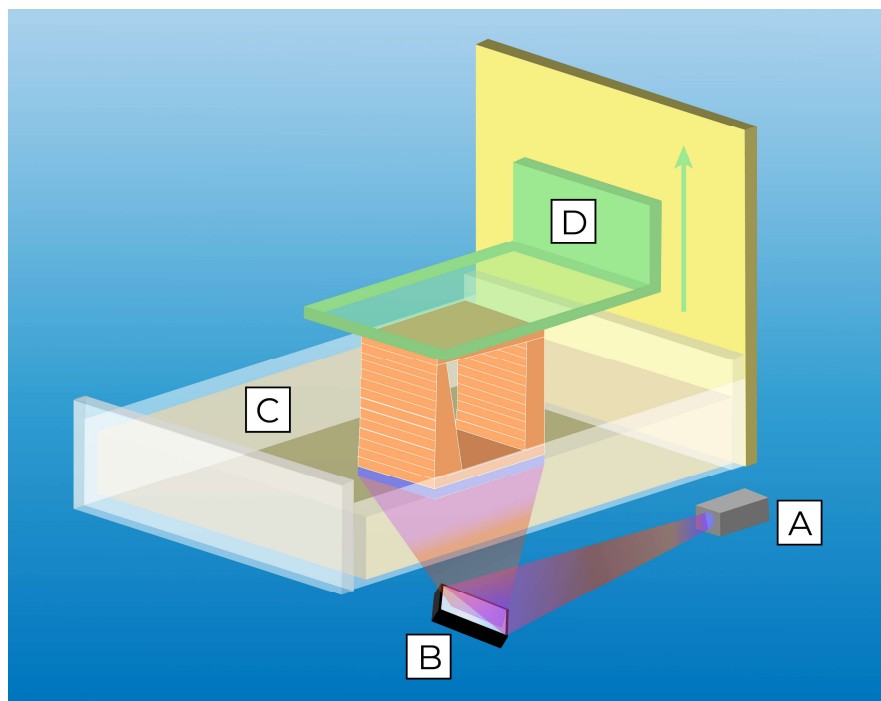

**Figure 5. Digital Light Processing (DLP)** 3D printing, a type of SLA with primary components, (A) photopolymerizing light source, (B) micromirror array to precisely direct light for layer curing, (C) vat containing photopolymer resin, (D) vertically mobile build platform.

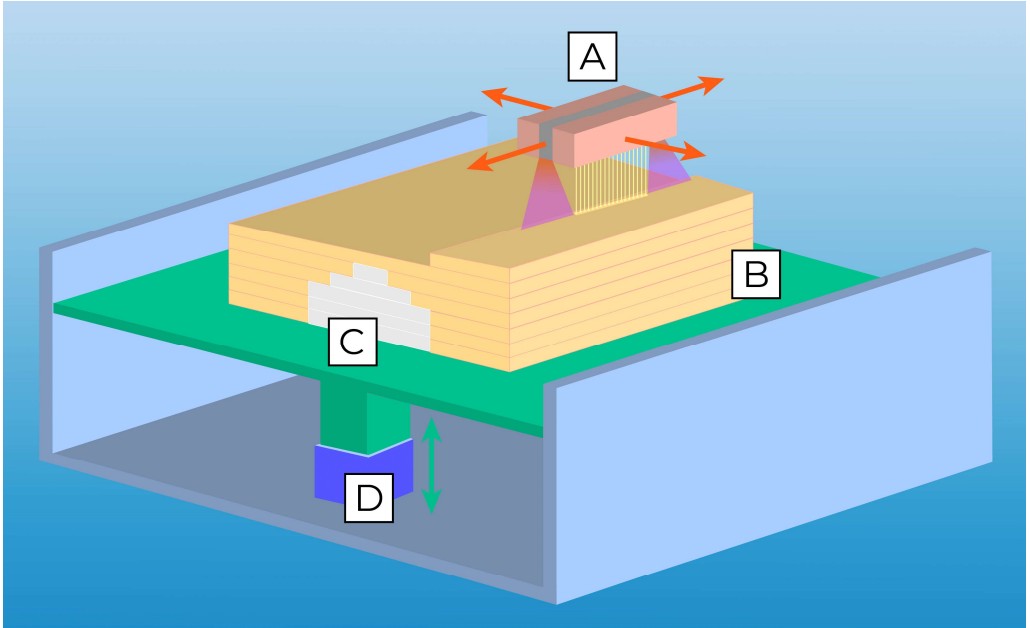

**Figure 6. Material jetting process** with primary components, (A) Inkjet head with multiple deposition nozzles and integrated UV light source, (B) layers of build material, (C) layers of support material, (D) build platform that moves vertically to accommodate each new layer.

Limitations of material jetting include the extensive support structures required, which occupy most of the non-product build volume, and high equipment cost [11]. Furthermore,

the high resolution achieved with thin droplet layers comes at the cost of slow printing times, especially in the vertical direction. However, the notable advantage of material jetting is the ability to build 3D multi-material and multicolor structures by combining different inkjet heads [11]. This is not easily achievable with other printing techniques, such as SLA or SLS, which are limited to a single material per print. Material jetting can also print both soft and hard photopolymers to create texture variety, which is limited when using other techniques [23]. Free-standing objects consisting of several materials with different optical or mechanical properties can now be manufactured by material jetting alone without an additional assembly step. The features of color reproduction, resolution, and material flexibility make material jetting printers suitable for applications requiring appearance accuracy, such as anatomical models and ocular prosthetics [24].

## 4. Applications of 3D Printing in Ophthalmology

### 4.1. Three-Dimensional Printing in Ophthalmic Implants and Prosthetics

Occasionally, patients who have suffered a serious orbito–ocular injury, including fractures, eye cancers, or trauma, necessitate the surgical extraction of the affected eye(s) [25]. In that regard, three surgical interventions may be performed, depending on the indication. First, evisceration is a procedure that entails the removal of the viscera of the eye while maintaining the extraocular muscles, Tenon's capsule, scleral coat, and optic nerve intact [26]. It is most indicated in cases of serious intraocular infection, causing pain and blindness [26]. Enucleation, on the other hand, implicates the surgical extraction of the complete eyeball by cleaving the optic nerve and the extraocular muscles, and it is performed to treat malignant ocular neoplasms, as well as irreversible traumas [26]. Finally, exenteration is considered the most invasive of the three surgeries, involving the eradication of all orbital material, including bone tissue, for the definitive treatment of advanced malignancies [26]. Following one of these operations, an orbital implant will be fitted and placed into the eyeless cavity to restore structure and adequate volume, followed by a visual prosthesis above the graft to achieve the look of a normal eye. In the traditional method, the surgeon establishes the approximate size of the orbital implant prior to surgery visually, based on the estimations of the patient's fracture site anatomy, combined with computed tomography-measured orbital volume [27]. However, the imprecisions of this technique can increase the risk of the implant not appropriately fitting the anatomy of the fracture, which can, in turn, cause ophthalmic complications, such as enophthalmos, diplopia, and displacement of the implant [28]. With the emergence of three-dimensional printing technology to create templates, customization of the implants in terms of size, shape, and contour can be more accurately achieved. In fact, 3D printing permits the creation of customized implants precisely fitting the fracture site or eye socket to better treat blowout fractures and congenital abnormalities, as well as aid evisceration and enucleation procedures. A 2018 study by Kang et al. described the review of 11 patients who underwent orbital wall reconstruction for orbital floor and medial wall fractures with the assistance of custom 3D-printed orbital implant templates. The templates were used per-operatively to shape implants before their insertion into the fracture site [28]. Quantitative analysis of patient outcomes was based on the CT measurements of the volume of orbital tissue confined in the bony orbit. This analysis revealed that all 11 patients presented no postoperative complications, and a statistically significant reduction was noted between the pre-operative and postoperative orbital tissue volumes of the affected orbit ($24.00 \pm 1.74$ vs. $22.31 \pm 1.90 \, cm^3$; $p = 0.003$) [28]. In contrast, the contralateral unaffected orbit and the reconstructed affected orbit had similar volumes ($22.01 \pm 1.60$ vs. $22.31 \pm 1.90 \, cm^3$; $p = 0.182$), demonstrating successful fracture repair [28]. Kormann et al. (2019) evaluated the biocompatibility of 3D-printed spherical orbital implants made of photocurable resin in 10 patients who underwent evisceration of painful blind eyes. To do so, they measured systemic toxicity by comparing serum biochemical markers before surgery and at 12 months after. Local toxicity was evaluated by assessing the signs of socket inflammation one month postoperatively, as well as changes in implant size on CT scans at two and 12 months after

surgery [29]. None of the ten patients presented signs of infection, inflammation, exposure, or extrusion of the implant, and no changes in implant size were revealed on computed tomography imaging [29]. Thus, this phase-1 clinical study attested to the biocompatibility of photocurable resin for SLA 3D-printed human orbital implants.

Ocular prostheses are custom-made eye models that can be used for cosmetic rehabilitation in individuals who were either born with a congenital eye abnormality or who have had their eyes removed as part of the treatment of various ocular diseases [30,31]. Many factors must be considered to create a realistic and symmetric prosthesis for the anophthalmic patient, which include the position, size, contour, and color of the device, as well as its weight, comfort, cosmesis, and motility [31,32]. Traditional methods of creating a customized prosthesis, which have remained relatively unchanged over the past century, are labor-intensive, time-consuming, and expensive, generally requiring the skills of an experienced ocularist or craftsman to hand-paint the iris and sclera [30]. In comparison, three-dimensional printing technology allows the fabrication of a high-quality, customized design based on the patient's eye anatomy in a significantly shorter time than for a conventional hand-made prosthesis [30,31]. In 2016, Ruiters et al. successfully 3D printed and fitted a patient-specific prosthesis for a 68-year-old male with anophthalmia secondary to evisceration surgery [33]. In this case, a digital 3D model of the patient's anophthalmic cavity was obtained using a CT scan, which is different from the traditional method of creating a mold by injecting impression material into the patient's eye socket. The 3D-printed prosthesis was dimensionally accurate but lacked color, so postprocessing had to be performed to add iris and scleral characteristics [33]. Likewise, in Alam et al.'s 2017 study, a white artificial eye was created using computer-aided design (CAD) and rapid 3D printing based on a CT scan of a wax model of two patients' orbits [32]. It was compared with a conventional custom-made prosthesis (CMP), and the CAD prosthesis was found to necessitate much lesser manufacturing time (2.5 h versus 10 h for the CMP). It weighed less (2.9 g compared to 4.4 g for the CMP), and it was subjectively more comfortable for both patients [32]. Kim et al. (2021) have also demonstrated a sublimation transfer printing technique that can reproduce the appearance of the contralateral healthy eye on a printed prosthetic without the need for manual painting [34].

Another form of non-invasive ocular prosthetics that can be three-dimensionally printed consists of eyelid crutches for the treatment of ptosis. Blepharoptosis can be quite debilitating for patients, especially in terms of vision deterioration and eye dryness caused by the difficulty in completely closing the affected eye. Despite surgical options to correct myopathy-induced eyelid drooping, there is a potential risk of recurrence after surgery [35]. Sun et al. (2018) reported using 3D printing to design and fabricate custom-fit crutches as an alternative and inexpensive therapeutic option for recurring ptosis. Not only were the printed eyelid crutches more affordable to fabricate than standard ones, but they could be easily removed and adjusted as well. Furthermore, after five months of usage, patients' vision was reported to have improved, and proper eye closure was possible [35]. Therefore, large-scale 3D printing and adoption of these devices can be contemplated to increase accessibility for patients struggling with ptosis in the near future.

Finally, macular buckles are surgically implanted devices that are used to treat an uncommon complication of myopia called myopic foveoschisis, which can increase the risk of retinal detachment and subsequent visual impairment [36,37]. Despite these devices being available in different shapes and sizes, their fit is frequently not perfect because of their generic structure. To remedy these issues, Pappas et al. (2020) created a custom macular buckle by 3D printing biocompatible polymers, specifically polyether ether ketone (PEEK), based on the exact 3D geometry of a patient's eye from CT imaging [38]. This customized device has the potential to decrease the complications associated with surgical intervention by minimizing the manipulation of extraocular muscles, sclera, and blood vessels by the surgeon. However, it was determined that the mechanical durability of the 3D-printed parts of the buckle was suboptimal in comparison to the pieces that were injection molded. Thus, it will be important to optimize the device's strength in upcoming

studies, and alternatives to radiation imaging, such as laser scanning microscopy or B-scan ocular ultrasound, will have to be explored to increase the safety of this technique [38].

In summary, there is evidently increasing promise for the widespread production of orbital implants and ocular prostheses using three-dimensional printing technology. Nevertheless, some limitations and challenges need to be addressed. First, even with the use of 3D printing, there are manual tasks that are still necessary to create customized ocular prostheses for patients, which can be time-consuming and make it only a semi-automated process [31]. These may include the impression mold of the patient's anophthalmic cavity, as well as the painting and polishing of the prosthesis [31]. In most studies, an ocularist was also needed to paint the external eye anatomy. However, there has been an instance where a photograph of the pupil, iris, and conjunctival blood vessels of the contralateral normal eye of the patient was taken using a slit lamp and subsequently printed on transfer paper, which was then transferred to the 3D printed prosthesis by sublimation [31]. This entire process added approximately an hour in total to the 3D manufacturing of the prosthesis, in contrast to the several hours required by a skilled ocularist to reproduce the iris and blood vessels manually. Another limitation is the cost of the printing itself. Direct 3D printing of titanium implants for orbital reconstruction can cost thousands of USD per implant [39]. However, the creation of orbital templates to shape standard implants, along with the continued maturation of 3D printing technologies, can significantly reduce costs while maintaining satisfactory results for patients [28]. Moreover, additional studies and clinical trials with larger sample sizes, as well as longer follow-up intervals, are required to corroborate the efficacy, cost reduction, cosmetic outcome, and patient comfort associated with 3D-printed ocular prostheses and orbital implants [28,31].

### 4.2. Educational and Anatomical Models

Conventional images of ocular anatomy may fail to contain sufficient detail and dimensional information for optimal teaching of medical students and residents. Three-dimensional printed anatomical models can recreate the detailed structure of the eye and its intricacies to enhance ophthalmology education. The orbit is one of the most complicated anatomical regions in the human body, with a high degree of individual variation. Despite this, 3D printed models of the orbit visualizing both bony structure and soft tissue can provide ocular anatomy details that improve learning, compared to textbook and computer-based learning methods [40]. In a meta-analysis of studies evaluating 3D printed models for anatomy education, Ye et al. found that 3D printed models resulted in faster answering time ($-0.61$, $p < 0.05$) and greater response accuracy among students in comparison to conventional methods such as cadaver and 2D learning [5].

A different randomized study by Wu et al. found educational value in a 3D-printed ocular model for teaching ophthalmoscopy to medical students [41]. The researchers randomly assigned 92 medical students to either a model-assisted training group, where students practiced on the simulated eye and with peers, or a traditional training group, where students only practiced with peers. After equal training time, both groups were assessed on their ability to see the fundus and determine the cup-to-disk ratio in patients. In the model-assisted training group, 43/46 (93.48%) students correctly determined the ratio, while in the control group, only 21/46 (45.65%) students saw the fundus and determined the correct ratio. The significant difference in skill acquisition suggests value in 3D printed ocular models for facilitating student learning before progressing to real patients. However, the study was non-blind due to the nature of the intervention, which may have mildly influenced student performance [41]. Although 3D printed anatomical models have benefited undergraduate medical learners, the literature has shown inconclusive benefits for resident physician learners and no statistical difference compared to training with 3D visual imaging [42].

However, 3D printed models remain a promising alternative to cadaveric specimens due to greater availability and reduced cost, especially for learners in geographically isolated areas [40]. They also offer greater anatomical reproducibility and can flexibly simulate

diverse pathologies processed from CT or MRI images. The non-perishable nature of 3D printed models makes them further suited for cost-effectively teaching specific and complex conditions [7,43]. A systematic review of 15 randomized trials evaluating 3D models for anatomy education found that models enhance knowledge acquisition, and most students are interested in utilizing 3D systems for learning compared to traditional methods [44]. In addition to education, 3D printing can be used for advanced visualization and diagnosis of ophthalmic conditions. Maloca et al. utilized optical coherence tomography (OCT) to image the architecture of choroidal vessels and tumors. These imaging data were then used to create 3D choroid models with FDM and SLA printing that illustrated the interactions of tumors with the vascular network of the choroid [45]. As choroidal tumors can disrupt the delicate vascular supply of the retina, 3D models that visualize and localize tumors can aid clinicians in diagnosis and treatment planning. Limitations regarding 3D printed anatomical models include the pliability and texture of many printable materials not being representative of real specimens. It is particularly challenging to replicate the anisotropic and viscoelastic variability of anatomy consisting of multiple tissues [23]. Finally, to produce accurate 3D anatomical models, models must be based on high-quality scans of specimens or prosections to ensure educational value [40].

### 4.3. Surgical Planning and Training

Surgical planning in ophthalmology is assisted by the ability of 3D-printed models to visualize patient anatomy and simulate operative procedures. Three-dimensional printed surgical simulators can provide visualized and tactile surgical training for procedures such as orbital surgery and keratoplasty. Notably, training with fracture models has been shown to enhance performance during operations and improve patient-reported functional outcomes [46]. A study by Famery et al. examined the performance of surgeons on Descemet membrane endothelial keratoplasty (DMEK) using a 3D printed platform. Human donor corneas were mounted on artificial anterior chambers with a 3D-printed iris, which allowed the adjustment of pupil size and anterior chamber depth for accurate simulation and modifiable surgical difficulty [8]. The realistic model enabled surgeons to practice all unfolding techniques that would be used in real surgery, and all surgeons, including beginners, completed the simulation with well-oriented grafts verified by OCT. Thus, the model demonstrates the value of 3D printing for surgical training with the ability to accurately simulate ophthalmic surgery.

Furthermore, 3D-printed surgical guides can be leveraged to benefit operative procedures directly. Fan et al. compared the clinical success of 3D-assisted orbital reconstructions compared to traditional surgery with a study of 56 patients [47]. The 3D-assisted technique utilized orbital models true to patients' anatomy, which allowed surgeons to conduct pre-operative planning. Surgeons also used the printed template to shape polyethylene-titanium mesh to better fit the patient during surgery. Compared to the control group, which underwent reconstruction without 3D printing, the 3D group achieved significantly shorter operating time ($75.34 \pm 15.68$ min vs. $95.37 \pm 22.19$ min; $p < 0.05$). Furthermore, the postoperative clinical results were superior in the 3D group, with significantly lower enophthalmos and a lower percentage of superior sulcus deformity [47]. Similar studies have printed custom 3D models of patients' orbits mapped from CT imaging and used these models to create pre-shaped implants for orbital fracture reconstruction [48,49]. Specifically, a titanium mesh was shaped and cut to size for patients according to their orbital anatomy, as represented in PolyJet-printed and resin 3D models. The use of the model reduced operation time, improved enophthalmos, and contributed to successful treatment outcomes while being relatively inexpensive to implement [48,49]. In addition to helping shape implants, 3D printing can directly create implantable polycaprolactone (PCL) mesh for use in the treatment of orbital wall fractures. A retrospective review of patient cases has shown that 3D-printed biodegradable PCL mesh enables ideal repair of orbital wall fractures with reliable stabilization and a low complication rate [50]. Furthermore, 3D-printed models of eyes with intraocular tumors have demonstrated utility in guiding

radiosurgery by enabling physicians to visualize tumor location physically and design more accurate stereotactic radiation therapy [13].

Challenges identified for 3D printing in surgical planning include the length of time required to design and print models (10–14 h) and the increased logistical complexity of treatment [48]. Overall, the long time required to image patient anatomy and print physical models makes 3D surgical preplanning less feasible for highly time-sensitive surgeries and emergent procedures. However, these are not common in ophthalmology, which enables the field to uniquely harness the benefits of 3D-printed surgical guides and planning models. A key advantage of 3D printers for surgical planning is the ability to visualize precise patient anatomy and diverse surgical cases using a single device. Applying 3D printing to aid surgeries can also enhance traditional procedures and provide templates to shape implantable devices [47]. The studies reviewed show that surgical preplanning models improve procedural readiness among surgeons and improve patient outcomes [48]. Three-dimensional printed simulators also present a unique opportunity for assessing and maintaining surgical competence for students and physicians in a controlled environment [8]. Finally, by demonstrating pathological conditions and surgical procedures on realistic models, patients can better understand their own conditions, which involves them in their care and helps build patient-provider trust. Although 3D rapid prototyping in ophthalmic surgery is promising, continued innovation in materials that accurately resemble ocular tissues is needed to allow for more realistic surgical simulation going forward.

### 4.4. Drug-Delivery Systems and 4D Printing

Three-dimensional printing technology has evolved considerably and rapidly in the last decade, and novel derivatives of 3D printing are currently in the works. For instance, four-dimensional printing, which encompasses time, is surfacing as a manufacturing method for medical materials and devices [51]. This novel technology empowers biomaterials to change their physical and functional properties over time, which promises to advance tissue engineering and enable new drug-delivery platforms [52]. For example, 4D-printed biomaterials can modify their structure in response to changes in temperature, pH, and ion concentrations, even after printing. These biomaterials can further undergo modifications to their function as the cells they contain mature [51]. Because of these characteristics, designs produced using 4D printing are designated as smart materials [53]. In healthcare, materials able to adapt their properties, functionality, and shape as a function of time have expanded to implants, targeted drug delivery, and complex surgery.

### 4.4.1. Drug-Eluting Implants

One potential 4D printing application in ophthalmology involves the use of 4D-printed hydrogel-based microneedles as a drug-delivery system that reacts to environmental stimuli [52]. Microneedles can be used as a simple, minimally invasive drug-delivery procedure with very little pain sensation [54–56]. These devices were classically used for the transdermal administration of different pharmaceutical agents, but with the impressive recent developments in the field of microtechnology, some studies have shown great promise for their use in the treatment of ophthalmic diseases [56,57]. The shape of the microneedles used in the 4D printing process can change if they are dissolved, under pressure, or cured with UV rays, which benefits the utility of precision drug-eluting implants [52]. Notably, implants that include intraocular pressure-responsive biomaterials can release IOP-lowering drugs at controlled times to treat glaucoma [51]. These 4D drug-eluting implants could be an alternative to topical eye drops, which include various hypotensive antiglaucoma agents [58]. These locally acting medications, which act by decreasing the production of aqueous humor or by increasing its drainage through the trabecular meshwork and the uveoscleral outflow, have their limiting factors [58]. They show low patient compliance, attributable to difficulties in their administration and ocular irritation or discomfort, particularly in the elderly [59]. Other limitations include their brief therapeutic time and less than 5% bioavailability, which is explained by various precorneal factors, such as blinking,

solution drainage, and tear film clearance [60]. Furthermore, the anatomical barriers of the cornea, conjunctiva, and sclera reduce drug absorption, necessitating frequent high-concentration doses of eye drops [61,62]. Therefore, with a 4D-printed implant releasing an antiglaucoma agent inside the eye, these limiting factors could be compensated.

Another promising therapeutic application of drug-delivery systems is for the treatment of retinal vascular diseases. Currently, the most widespread clinical therapy for diabetic retinopathy and its potential complication, diabetic macular edema (DME), as well as wet age-related macular degeneration (AMD) and macular edema secondary to retinal vein occlusion (RVO), consists of repeated intravitreal injections of an anti-vascular endothelial growth factor, or anti-VEGF, medication [63–65]. Anti-VEGF agents, such as bevacizumab, inhibit a crucial growth factor in the pathogenesis of neovascularization, the process in which new immature blood vessels are formed [59,63]. This pathophysiological characteristic is seen in proliferative diabetic retinopathy, where hyperglycemia promotes retinal neovascularization by regulating the synthesis of VEGF and PEDF (pigment epithelium-derived factor). In wet AMD, angiogenesis instead occurs in the choroid layer behind the retina of the eye [59,63]. These capillaries are very fragile and can easily leak exudate, which can precipitate vitreous or subretinal hemorrhage, fibrosis, and tractional retinal detachment. In the worst-case scenario, capillary bleeding can cause irreversible retina damage, vision impairment, and even blindness [59,64]. Consequently, minimizing this potentially harmful neovascular process is of greatest importance in these retinal and macular diseases, and anti-VEGF injections have proven to be the gold standard in preserving and improving visual acuity for disease of the retina [63,65]. However, to deliver adequate quantities of an anti-VEGF medication to the posterior segment of ocular tissues and to counterbalance the rapid clearing of the medication from the vitreous body, frequent intravitreal injections are necessary, especially in the first few months of therapy [59]. These recurring treatments, in combination with frequent visits to an ophthalmologist's office every four to eight weeks, can constitute a significant physical, emotional, and economic burden not only on patients but also on their caretakers, as well as on healthcare professionals [63,66–68]. Moreover, it was demonstrated that these repeated injections can increase the risk of retinal detachment, hemorrhage, and intraocular inflammation [59,63,69]. To avoid these potential complications, Won et al. (2020) developed a drug-loaded rod, also called a drug rod, using a flexible coaxial 3D printing technique, which was implanted in rat vitreous using a minimally invasive small-gauge needle and delivered bevacizumab and dexamethasone in a time-controlled manner into the vitreal cavity [63]. The drug rod incorporated an external shell that was 3D printed using polycaprolactone and bevacizumab (PCL-BEV), and the interior core contained an infusion of alginate and dexamethasone (ALG-DEX). Coaxial printing was achieved with a multiple-head 3D bioprinter and a set of coaxial nozzles containing numerous combinations of core/shell needles [63]. Specifically, the PCL-BEV ink, formed by the dilution of both substances in dichloromethane (DCM), was distributed in the shell needle of a coaxial nozzle, while a hydrogel was simultaneously released by the core needle of the same nozzle. The interior core ALG-DEX bioink was assembled by diffusing sodium alginate in deionized water and combining this solution with dexamethasone. During the printing process, the PCL-BEV shell rapidly solidified due to evaporation of the DCM solvent, and the hydrogel core was removed by deionized water and replaced by the administration of the ALG-DEX ink to form the drug rod. Subsequent in vitro and in vivo studies proved that the structural design and the biomaterials comprising the rod allowed the controlled release of both bevacizumab and dexamethasone, as well as extended their therapeutic duration, compared to the conventional intravitreal treatments. In fact, the drug rod was able to continuously deliver BEV for 60 days, in contrast to the injected BEV's 2-week half-life. Additionally, choroidal neovascularization was inhibited by the drug rod over a 4-week evaluation period in a rat model, whereas the intravitreal bevacizumab was able to suppress angiogenesis for only 2 weeks [63]. Therefore, not only is this technology able to reduce the side effects associated with intravitreal injections, but it can improve

compliance by increasing the drugs' release period, as well as making their administration more bearable for patients since the rod's implantation process is a much less invasive technique [63,70]. Nonetheless, more studies will be required to evaluate their safety for use in humans, as well as determine the most efficacious combinations of drugs, doses, routes, and drug-release patterns that will better stabilize degenerative retinal diseases while maintaining a minimal side-effect profile.

4.4.2. Drug-Eluting Contact Lenses

3D printed drug-eluting contact lenses are another novel technique that has the potential to revolutionize the treatment of various ocular conditions, including keratoconjunctivitis sicca, or dry-eye disease, age-related macular degeneration, and glaucoma [71]. In fact, these lenses are not only useful to correct visual acuity deficits and refractory errors but can also deliver medications in a controlled manner and offer greater bioavailability to the eye's surface compared to standard eye drops [59,72]. When a contact lens is deposited onto the cornea, the tear film is divided into two components: the pre-lens tear film (PLTF), in which drugs are absorbed by the conjunctiva or gain access to the systemic circulation by entering the canaliculi, and the post-lens tear film (POLTF), where medications diffuse through the cornea using a direct approach [59]. Drug-eluting contact lenses can be manufactured using 3D printing techniques such as FDM, as demonstrated by Mohamdeen et al. [73]. They fabricated lenses from a blend of ethylene-vinyl acetate copolymer (EVA) and polylactic acid (PLA) using hot melt extrusion. Integrated with the lens filament was timolol maleate (TML), a glaucoma medication that reduces intraocular fluid production. An EVA/PLA/TML ratio of 84:15:1 (wt./wt.) was found to be ideal for printability, lens integrity, and drug release. The 3D printed lens released loaded TML over 3 days but only eluted 35% of the total drug. The authors reason that sustained release was not achieved due to slow diffusion from the polymer matrix, and further work is needed to optimize drug release [73]. Methods for optimizing ocular drug delivery include integrating different nanocarriers into the lenses' composition, such as polymer nanoparticles, liposomes, micelles, and microemulsions [74]. These nanomaterials are also important not only to prevent the enzymatic degradation of the drug but also to minimize the possible medication leak during its storing and sterilization processes [71,75]. Factors that will require more consideration in the future to create safe and effective drug-delivery systems using contact lenses include biocompatibility, oxygen permeability, tensile strength, optical transparency, sterilization, and storage, without forgetting patient comfort [76,77]. Future development of smart and drug-eluting lenses leveraging 3D printing could offer a minimally invasive and safe route for ocular drug delivery [78].

*4.5. Four-Dimensional Orbital Implants*

An additional 4D printing prospect concerns the treatment of enophthalmic invagination. Enophthalmos is described as the posterior displacement of the normal-sized ocular globe within the orbit following an anteroposterior plane [79,80]. This relative shift can occur following orbital trauma or not, and it is corrected by filling the orbital volume with an implant, which in turn can reinstate facial symmetry [80]. Unfortunately, the current implant devices that are used lack precision and capability to fill the increased volume, and they necessitate large surgical incisions to be correctly implanted [79]. Shape memory polymers (SMPs) are printable stimuli-responsive smart materials and can present in different temporary and permanent shapes when exposed to heat, electrical fields, light, magnetic fields, and solutions [79,81–83]. Shape memory polyurethane specifically has an adjustable transition temperature that is determined by the melting temperature of its soft segment, and its firmer segment dictates its permanent structure [84,85]. It also possesses satisfactory mechanical characteristics, antithrombotic properties, and biocompatibility that make it a safe material for the production of personalized ophthalmic implants in the near future [79]. Deng et al. (2022) created an orbital stent based on CT reconstruction technology, and 4D printed a shape memory polyurethane composite to treat enophthalmos [79]. In its compressed temporary

state, the stent was implanted in a minimally invasive fashion into rabbits before thermal stimulation enabled the assumption of its permanent shape. The volume filling ability was nearly 150% greater compared to two commercially available implants, which included Medpor, made of porous polyethylene, and absorbable plates [79]. Thus, printed stents leveraging shape-changing materials can enable precise treatment of enophthalmos.

*4.6. Adaptive Optics*

Adaptive optics refers to a non-invasive technique that corrects optical aberrations using deformable mirrors, which can be applied to the eye and accurately depict the retina's cells [86,87]. This concept was first proposed in 1953 by American astrophysicist Horace Babcock to refine the telescopic images of distant stars, which lacked precision and clarity because of the optical deviations caused by Earth's atmospheric turbulence [86,87]. Likewise, with the eye's anatomy being very complex and made of different tissues, the differences in the refractive indexes of these ocular tissues create wavefront chromatic and monochromatic aberrations when light rays exit the eye [87]. Monochromatic aberrations are further classified as being low-order or high-order. Lower-order aberrations include refractive errors, such as myopia and hypermetropia, as well as astigmatism, and despite them being of greater importance and much more prevalent, they are easily corrected with spherical and cylindrical lenses, respectively [87,88]. On the other hand, higher-order aberrations, like keratoconus, are far less common but are more arduous to correct [88]. In 1997, Liang et al. were able to put together a fundus camera, combined with a Shack-Hartmann wavefront sensor (SHWS) and a deformable mirror, to produce high-quality images of the retina at its cellular level, specifically the cone photoreceptors, by overcoming the higher-order monochromatic aberrations [89]. This was the first application of adaptive optics in ophthalmology, and it paved the way for numerous studies assessing the different retinal components in vivo, such as photoreceptors, retinal pigment epithelial cells, and microvascular anomalies [86]. An in-depth examination of retinal cells and anomalies can provide a better understanding of the diseases affecting the retina, as well as help in their diagnosis before substantial damage occurs [87]. Thus, existing treatments for retinal pathologies could be administered as a preventive measure, and new therapeutic modalities could be developed to better control or even stop the progression of these diseases. Adaptive optics can also be combined with other retinal imaging techniques, such as flood illumination ophthalmoscopy (FIO), scanning laser ophthalmoscopy (SLO), optical coherence tomography (OCT), fundus fluorescein angiography (FFA), and indocyanine green angiography (ICG), and complement their findings [86]. There are four main components to the standard adaptive optics equipment:

(1) A wavefront sensor to qualify and quantify the optical aberrations in the light reflected by the eye;
(2) A deformable mirror to correct the identified abnormalities;
(3) A control system to calculate the necessary correction amount and to provide feedback, and;
(4) A processing device to create an image based on the corrected waveform.

In terms of 3D and 4D printing, López-Valdeolivas et al. (2017) described the 4D manufacture of a liquid crystalline elastomer (LCE)-embedded polydimethylsiloxane (PDMS) actuator that could be used for adaptive optics, owing to this material's flexibility, effortless handling, translucency, low weight, absence of toxicity and cost-effectiveness [90]. Hammer et al. (2019) created a biomimetic phantom that corresponded to the human retina to evaluate the performance of adaptive optics. The retinal model mimicked the photoreceptor mosaic, respecting the arrangement and the size of cells, and its cone photoreceptors were 3D fabricated using the two-photon polymerization technique [91]. This model eye was designed to allow imaging with SLO and OCT with the potential to help in the evaluation of AO device functioning. In the future, the retinal images generated from AO could be 3D-printed to be used as educational models for surgical planning purposes, and eventually, 3D bioprinting of retinal cells could also be achieved for possible transplantation. Nonetheless, there are challenges associated with the use of 3D-printed adaptive optics in a

clinical setting. This includes its very high purchase price and the potential difficulty in obtaining satisfactory quality images, especially in eyes that present diverse abnormalities, such as dryness, cataracts, corneal scars, vitreous debris, or involuntary ocular movements like nystagmus [86]. Other limitations concern the very narrow zone that can be imaged at a time, meaning that some areas of retinal pathology could be omitted, and the very time-consuming and complex analysis of the images [86].

In short, there are many novel technologies currently being studied in ophthalmology, which include different 3D and 4D printed drug-delivery systems, such as implants, shape memory polymers, and adaptive optics imaging. All these innovations have the potential to aid the treatment of ocular diseases, but these are not without limitations and side effects. Therefore, additional studies will be necessary in the near future to attest to their safety for human use.

## 5. Bioprinting

3D bioprinting is an emerging technology that leverages the unique advantages of 3D printing for tissue engineering. It involves printing biological systems using bioinks rather than plastics or composites. The fluid bioinks used in bioprinting contain live cells, growth factors, and hydrogels that can be precisely deposited in layers to simulate functional tissues (Figure 7), such as cartilage, vasculature, or even ocular structures like the cornea [92]. Bioprinted constructs aim to achieve biomimicry of their target tissue's physiological functions by embedding cells within hydrogel scaffolds that resemble natural tissue. They then leverage the ability of cellular progenitors within bioink to autonomously self-organize into tissues [93]. Thus, bioprinting has substantial potential in regenerative medicine through the creation of substitute tissues and organs for patients. With the increasing need for donor organs and expanding waitlists globally, this technology is needed now more than ever. Bioprinting products can be designed to fit patients' individual anatomy using imaging-guided CAD, and if manufactured with patients' own cells, can reduce the risk of immune rejection seen with allogeneic transplants. Furthermore, bioprinted tissues can be used to study the pathogenesis of diseases such as cancer with in vitro tumor models printed using native tumor cells. As these bioprinted models can mimic the unique microenvironment of diseases, they serve as a valuable platform for testing and advancing new therapeutics [94]. The capabilities of bioprinting open avenues for the fabrication of ocular tissues, including corneal, retinal, and conjunctival models, for research and potential transplantation in the future. To provide a background, the techniques used in bioprinting are similar to traditional 3D printing, and an overview of the main domains will be briefly elaborated.

### 5.1. Extrusion-Based Bioprinting

Extrusion-based bioprinting techniques deposit bioink in continuous filaments through a mobile microscale nozzle to produce defined structures. In this manner, they rely on similar technology to FDM 3D printing, which also utilizes material extrusion. The extrusion of bioink during printing is finely controlled via mechanical pressure exerted by a piston, pneumatic pressure, or through a rotary screw (Figure 8), depending on the bioprinter design [18,95]. A microvalve system can further be utilized at the nozzle for fine control of bioink deposition. Following extrusion, the bioink must be stabilized to maintain its shape, often using hydrogel crosslinking agents [96]. This form of bioprinting affords flexibility in bioink formulation, including pastes, dispersions, and additive-containing solutions that enable fabrication with the high cell density needed for complex and vascularized tissues [97]. Extrusion-based printing is limited by a relatively lower resolution compared to stereolithography, droplet-based, and laser-based bioprinting (Table 2).

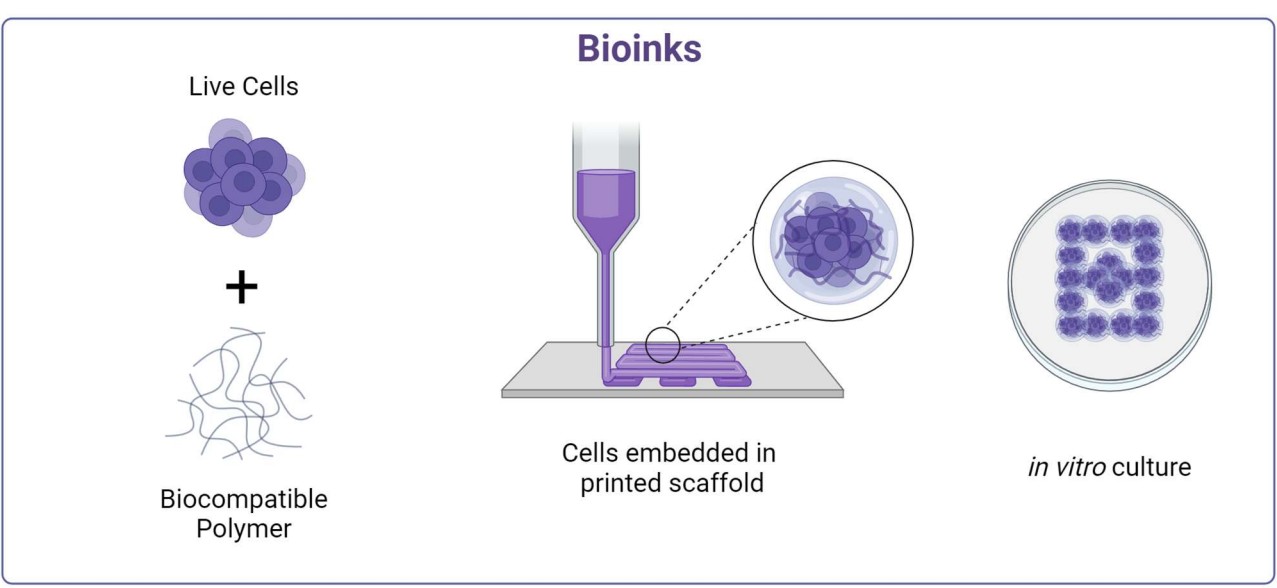

**Figure 7.** Bioinks combine cells with structural materials such as uncrosslinked hydrogel polymer. Following printing, polymers undergo crosslinking to maintain the shape of the scaffold, and embedded cells can be grown in culture.

### 5.2. Droplet-Based Bioprinting

Droplet-based bioprinting is fundamentally similar to material jetting 3D printing and is a process where discrete volumes of bioink are selectively deposited onto a build platform to assemble the biological structure layer by layer. The most common technology for droplet deposition is ink jetting, where piezoelectric or thermal expansion generates pulsed pressure to expel precise droplets of bioink onto a culture dish or hydrogel substrate [95,98]. Droplet-based technologies can use low-viscosity bioinks, have fast fabrication speed, and achieve high printing resolution, which is useful for intricate tissues with multiple cell types [99]. However, inkjet bioprinters are limited by lower printable cell densities and are not ideal for large vertical structures [96].

### 5.3. Laser-Assisted Bioprinting

Laser-induced forward transfer (LIFT) is a high-resolution deposition method that directs light energy toward a ribbon consisting of an absorbing layer and bioink. The laser heats and expands the absorbing layer, generating a localized bubble of high pressure that displaces bioink and cells to the hydrogel-coated substrate [18]. This mechanism can achieve a printing resolution as small as 10 μm with biomaterials in solid or liquid phase [96,100]. Drawbacks include the high cost and risk of thermal damage to cells from laser exposure. LIFT systems can minimize this risk of laser irritation and maintain high cell viability by utilizing bioinks with higher viscosities and depositing the cells onto thicker hydrogel films as substrates [96,101].

### 5.4. Stereolithography Bioprinting

Stereolithography (SLA) leverages a directed light source to selectively cure a liquid cell–hydrogel suspension into a 3D structure. The technique relies on the targeted crosslinking of hydrogels containing light-sensitive photopolymers such as polyethylene glycol–diacrylate (PEGDA) and gelatin-methacryloyl (GelMA) [18]. As with regular SLA and digital light processing 3D printing, the excellent accuracy and fast printing times are major advantages [102]. Challenges with SLA bioprinting include maintaining cell viability, as conventional UV laser curing damages cellular DNA [103]. However, recent advances have demonstrated the feasibility of using photoinitiators such as eosin Y to crosslink bioinks with visible light [18,104].

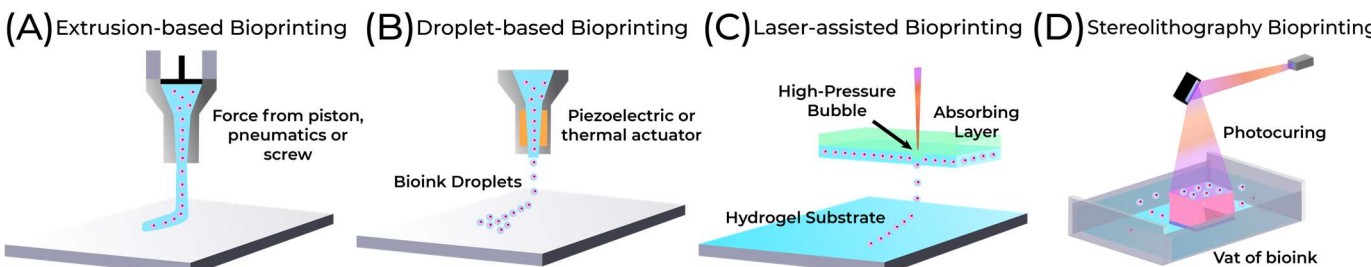

**Figure 8. Bioprinting Techniques** (**A**) Extrusion bioprinters use force created by a piston, pneumatics, or a screw to continuously extrude a liquid cell–hydrogel bioink. (**B**) Inkjet printers deposit small droplets of cells and hydrogel sequentially to build up tissues. (**C**) Laser-assisted bioprinting utilizes a laser to vaporize a region in the absorbing layer, forming a bubble that precisely displaces bioink towards the substrate. (**D**) Stereolithography uses a finely controlled light source to selectively crosslink hydrogels within bioinks to build a 3D structure layer by layer.

**Table 2.** Summary of Primary Bioprinting Techniques.

|  | **Extrusion** | **Droplet** | **Laser-Assisted** | **Stereolithography** |
|---|---|---|---|---|
| **Advantages** [9,96,102,105] | Biomaterial flexibility<br>High printable<br>cell densities | Ability to print<br>low-viscosity bioinks<br>Fast printing speed<br>High resolution | High resolution<br>Capable of printing<br>bioinks in liquid or<br>solid phase | Fast printing time<br>High resolution<br>Nozzle free, no shear stress<br>High cell viability with<br>visible light |
| **Limitations** [102,105,106] | Requires viscous bioinks | Limited capability for<br>vertical structures<br>Low cell densities | High cost<br>Risk of thermal<br>damage to cells | Risk of damage to cells if<br>using UV<br>Requires photopolymer bioink |
| **Resolution** [100,104,106] | Medium (100 μm) | High (50 μm) | Highest (~10 μm) | High (50 μm) |
| **Print Speed** [105,107,108] | Slow | Fast | Medium | Fast |
| **Supported Viscosities** [93,108–110] | 30 mPa/s to<br>above $6 \times 10^7$ mPa/s | 3.5 to 12 mPa/s | 1 to 300 mPa/s | 250–$1 \times 10^4$ mPa/s |
| **Cell density** [93] | High | Low | Medium | Medium |
| **Cell Viability** [104,111–114] | <90% | 80–95% | <85% | 85%–>90% |

## 6. Applications of Bioprinting in Ophthalmology

### 6.1. Cornea

The cornea is the transparent outermost layer of the anterior eye that transmits and focuses light entering the eye. Corneal health is essential for vision, and thus, impairment of the cornea from dystrophies, injurious stimuli, and bacterial infections is a cause of blindness for millions worldwide, according to the World Health Organization [115,116]. Notably, about 53% of the population does not have access to corneal transplantation, and in countries where it is available, there is a chronic shortage of donor corneas [117,118]. Overall, only 1 in 70 patients who require a cornea transplant have their needs met [117]. As an alternative, 3D corneal bioprinting has the potential to substantially increase accessibility to transplantation in patients with corneal blindness in the future. The cornea is a uniquely promising candidate for bioprinting as its relatively homogenous structure and avascularity enable it to be modeled with 3D printing technologies. It is composed of 3 layers: a superficial epithelium, a middle stromal layer, and a deep endothelium (Figure 9). The stromal layer comprises over 90% of the cornea and consists of an extracellular matrix (ECM) with embedded keratocyte cells that organize the ECM structure [119]. The homogeneity of the collagen fibrils in the stroma facilitates its printability with bioinks [120]. Likewise, the avascularity of the cornea reduces manufacturing complexity and lessens the likelihood of immune reaction toward bioprinted transplants [121].

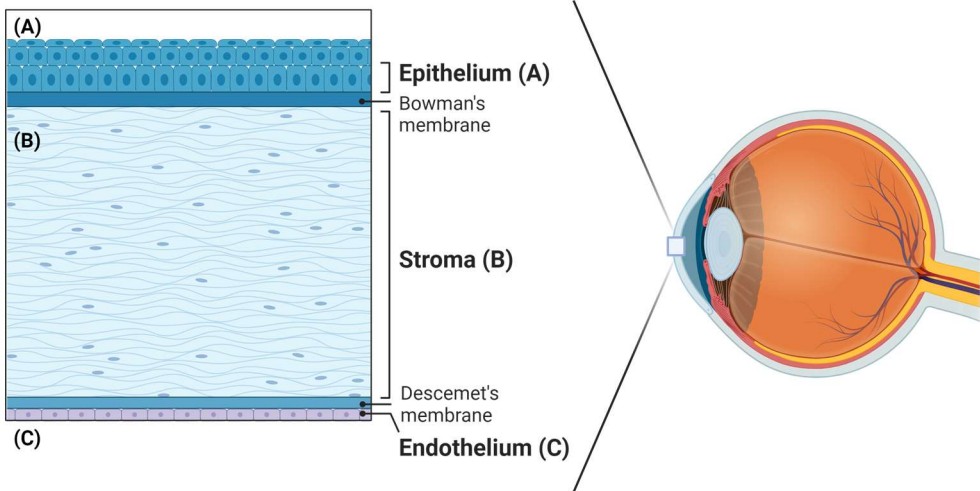

**Figure 9.** A diagram of the layers of the human cornea showing (A) epithelium, (B) stroma, and (C) endothelium. The acellular Bowman's and Descemet's membranes separate the stroma from the epithelium and endothelium, respectively.

Isaacson et al. printed corneal tissues resembling the human corneal stroma using pneumatic 3D extrusion bioprinting [122]. The bioink that built these corneal models was composed of methacrylated collagen and sodium alginate with encapsulated corneal keratocytes. Ideal printing occurred through a 200 μm nozzle, and a 3D printed support structure was used to provide corneal curvature. The printed structure was able to mimic the structure of the corneal stroma, with a collagen network resembling the ECM that served as a scaffold for live keratocytes. These keratocytes maintained high viability with >90% survival at 1 day post-printing and 83% at day 7, which suggests the suitability of collagen-alginate composite bioinks for bioprinted corneas [122]. This proof-of-concept cornea model achieved a proper shape and demonstrated transparency but lacked epithelial and endothelial layers. Future directions include supporting keratocyte maturation within the stromal matrix and attaining the higher transparency needed for a clinically suitable corneal graft [122,123]. Kim et al. worked towards achieving a bioprinted cornea with improved transparency by controlling shear stress at the extrusion nozzle to align collagen fibrils and influence keratocyte remodeling. Shear stress was varied by changing nozzle diameters, and prints from the 25 gauge nozzle were found to achieve an ideal transparent lattice-like microstructure after keratocyte remodeling in culture and in vivo when transplanted into rabbits. This was compared to a pipette-created cornea stroma with randomly oriented collagen fibrils. The printer-aligned samples showed superior optical transmittance in both culture and rabbit corneas after 28 days of remodeling. Thus, the study demonstrates that aligning collagen orientation with shear force direction contributes to transparency in bioprinted corneal transplants. It further shows the feasibility of in vivo transplantation of bioprinted corneal tissue with active keratocyte integration into rabbit corneas [124]. In addition to studies examining extrusion printing, droplet bioprinting was used by Campos et al. to produce a 3D stromal equivalent [125]. Leveraging a blended collagen-type I-agarose bioink with encapsulated keratocytes enabled the production of a dome-shaped corneal model. The embedded keratocytes were determined to maintain their native dendritic shape and phenotype after 7 days in a culture based on positive expression of keratocan and lumican biomarkers [125]. In summary, corneal stromal models with cellular viability have been created using extrusion- and droplet-based bioprinting. These may serve as foundations for future research on transplantable stroma tissue for patients suffering from corneal diseases.

Furthermore, laser-assisted (LIFT) bioprinting of corneal epithelium in addition to stroma has been demonstrated by Sorkio et al. [126]. The bioprinted cornea used human limbal epithelial stem cells as the source for the epithelium, and adipose-derived stem cells

comprised the stroma. Laser-printed tissue successfully formed a stratified epithelium and horizontally layered stroma with good cell survival and growth protein expression. The 3D bioprinted tissues were also implanted into porcine corneas, demonstrating host tissue integration and mechanical integrity after 7 days in culture [126]. Thus, 3D laser-assisted bioprinting can flexibly print with stem cells to replicate characteristics of the human cornea while maintaining functional properties.

Bioprinting of the corneal endothelium has also been shown by Kim and colleagues in 2018 [127]. Extrusion-based printing was used by the researchers to create two grafts of human corneal endothelial cells with and without overexpression of ribonuclease (RNase) 5, a protein that promotes cell survival. These grafts were transplanted into rabbit corneas, and both bioprinted endothelia helped restore corneal clarity with improved corneal thickness and edema compared to the control. Notably, the RNase 5 graft demonstrated expression of more phenotypical markers with higher cell population, growth velocity, and greater sodium-potassium pump expression than the regular graft. Thus, the results suggest that corneal endothelial cells can be genetically modified to achieve greater graft cellularity, which could benefit the successful integration of transplanted grafts [127].

The potential advantages of 3D bioprinting include the creation of personalized corneal implants with controllable structure and designed refractive ability. Research into printing corneas using a patient's own stem cells may also reduce the risk of immune rejection associated with donor corneas, thus improving long-term patient outcomes from transplantation [121]. Although the feasibility of bioprinted corneas has been demonstrated, further work is needed for clinical translation. This includes replicating the complex microstructure of the human corneal stroma to achieve optimal mechanical properties and transmittance. More in vivo research is needed to maintain cellular viability in printed corneas and biocompatible integration with host tissues.

### 6.2. Retina

The retina is the posterior-most layer of the eye, a visible extension of the central nervous system that enables phototransduction, the process that converts light energy from photons into electrical signals, which are subsequently interpreted as images in the brain [10,128]. No less than 130 million cells of over 60 different types, regrouped in distinct cell lines and three basic cell types, form the ten individual layers of the retina (Figure 10) [128–130]. Each cell has a particular role to play in generating vision (Table 3).

**Table 3.** Function of the different retinal cells.

| Basic Cell Types | Cell Lines | Main Function [129,131–135] |
|---|---|---|
| Photoreceptor cells | Rods | • Low-light, black and white vision<br>• ~95% of photoreceptors<br>• Concentrated in the retina's periphery, none located in the fovea |
| | Cones | • Color vision: detect either red (64%), green (32%), or blue light (2%)<br>• ~5% of photoreceptors<br>• Concentrated in the central area of the retina (fovea) |
| Neuronal cells | Retinal ganglion cells (RGCs) | • Retina's main output neuron<br>• Transmit both image (photoreceptor function) and non-image-forming information<br>• Receive both excitatory and inhibitory outputs from amacrine and bipolar cells<br>• Send axonal projections that meet in the optic disc |

**Table 3.** *Cont.*

| Basic Cell Types | Cell Lines | | Main Function [129,131–135] |
|---|---|---|---|
| Neuronal cells | Amacrine cells | | • Intermediate neurons that release primarily inhibitory neurotransmitters (GABA, glycine)<br>• Excitatory activity also possible<br>• Utility cell of the retina: many functions via microcircuits to detect different shades and movements of light in various directions<br>• Paracrine function (including dopamine release) |
| | Bipolar cells | | • Second-order long-projection neurons<br>• Receive visual inputs from photoreceptors<br>• Project to RGCs |
| | Horizontal cells | | • Modulate information transfer from bipolar cells and photoreceptors<br>• Inhibitory inputs via GABAergic interneurons |
| Glial cells | Microglia | | • Resident macrophages (main immune cells) |
| | Macroglia | Astrocytes | • Provide homeostatic and metabolic support to photoreceptors and neurons |
| | | Müller cells | • Ensure structural stability of the foveal tissue<br>• Improve light transmission to the photoreceptors |

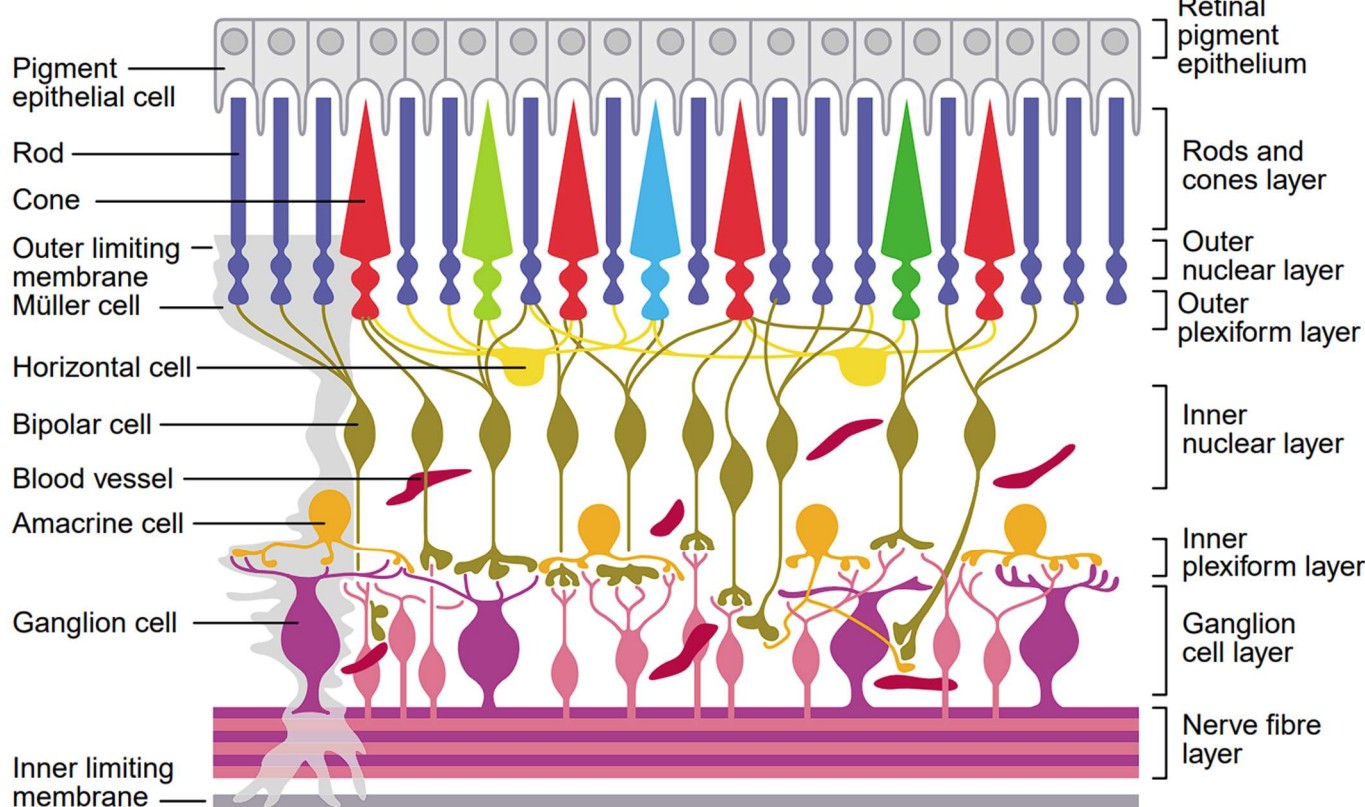

**Figure 10.** The cellular organization of the retina. Reprinted with permission [130].

The outer retina is delimited by the retinal pigment epithelium (RPE), which supplies growth factors and ensures the transport of nutrients, as well as the phagocytosis of the photoreceptors [136]. The RPE is followed by the choroid, the posterior segment of the uveal

tract [129]. The retina is the most metabolically active tissue in the human body and is thus vascularized by two different blood supplies: the outer layers are nourished by the choroid, whereas the inner layers are irrigated by branches of the ophthalmic artery [129,137].

The retina is undoubtedly complex, both structurally and functionally. Damage to retinal cells can result in the development of numerous eye diseases, leading to gradual vision loss. Any impairment to one or more retinal layers can lead to the degeneration of the photoreceptors and the subsequent atrophy of the retinal pigment epithelium. Some pathologies are caused by the loss of only one cell type, such as malfunctioning retinal ganglion cells seen in glaucoma and faulty photoreceptors associated with retinitis pigmentosa [138,139]. Others require a larger affected area of the retina, such as age-related macular degeneration, stemming from chronic inflammation of the macula, the retina's center [10]. To restore vision, all layers of the retina must be fully functional, meaning that the damaged parts must be regenerated [51]. To date, the grafting of new photoreceptors, progenitor cells, retinal sheets, and RPE cells, for example, has been attempted to reverse the many pathologies that can affect the retina [140–142]. However, once they were implanted, a substantial number of cells did not survive, and if they did, their structure and behavior were often atypical [143,144]. In subsequent studies, microfabricated scaffolds were used and were proven to deliver cells in a more controlled manner, but none possessed all the necessary structural properties or were able to encompass enough well-positioned and oriented cells [145–148]. Thus, 3D bioprinting has been suggested to remediate these issues, and a few studies have already demonstrated progress towards a viable retina using this technology.

First, Lorber et al. (2013) reported the successful inkjet 3D bioprinting of two types of adult rat central nervous system cells (CNS), retinal ganglion cells (RGC) and glia cells, with a piezoelectric printer [149]. It was determined that many cells remained in the nozzle, and consequently, lower concentrations of cells were able to be included in the printed scaffold. However, the survival, phenotype, and growth-stimulation characteristics of the cells were mostly preserved, even when exposed to shear stress, and their viability was deemed sufficient when compared to the non-printed control cells (69% vs. 78% for the glial cells, 74% vs. 69% for the retinal cells) [149]. To create their scaffold, Kador et al. (2016) used another bioprinting method, the thermal inkjet bioprinting technique, to deposit retinal ganglion cells on an electrospun matrix made of polylactic acid [150]. The survival rate of the cells was again satisfactory, and they preserved their electrophysiological properties relatively well. The cells' orientation was also assessed and deemed to be ameliorated compared to the control, with proper alignment for 72% of the axons and 49% of the dendrites, in comparison to 11% of the control cells [150].

To reproduce the retina's structure closely, Shi et al. (2018) used microvalve-based bioprinting to create a bi-layer structure. The first layer was composed of an RPE cell monolayer on an already formed PCL membrane to replicate the RPE, and the second layer was built from an alginate and Pluronic F-127 bioink containing photoreceptor cells [151]. During the printing process, the bioink and cells maintained their form and viability, with the cell numbers multiplying over two weeks and not seeming to undergo apoptosis [148,151]. Similarly, Wang et al. (2018) developed a two-layer scaffold by laser-assisted 3D bioprinting of a hyaluronic acid (HA)-based bioink. This bioink was chemically modified with methacrylation by glycidyl-hydroxyl reaction to produce a photopolymerizable hydrogel that resembled the human retina's stiffness [152]. The two layers differed in their thickness and cell composition, with the upper layer measuring 250 μm in thickness and containing fetal retinal progenitor cells (RPCs), whereas the lower layer of RPE cells, added to enhance the progenitor cells' differentiation, was 125 μm thick. The study's results showed that the survival of the RPCs was greater than 70% when incorporated in an environment similar to the native retina, proving that such conditions promote the maturation of the progenitor cells [152]. Masaeli et al. (2020) also covered a bioprinted RPE with a layer of photoreceptors using an inkjet bioprinting apparatus, but in contrast to the other mentioned studies, they did not include any carrier or scaffold material [153]. Three days following bioprinting, both layers of cells were correctly positioned one on top of the other, and

it was confirmed that the RPE layer was functional because human vascular endothelial growth factors (hVEGFs) were detected in substantial quantities. Thus, this indicated that they had created an acceptable in vitro retina model that could be used to study various sight-threatening retinal pathologies [153].

Thus far, the studies regarding 3D bioprinting of the retina have evaluated the following factors:

1. The cell viability after printing
2. The bioprinted scaffold's structure
3. The cells' orientation inside the scaffold
4. The cells' arrangement in various layers

Nevertheless, despite these promising results, further research will be necessary to achieve the goal of bioprinting whole and functional retinal tissues for grafting purposes in patients living with degenerative diseases of the retina. To do so, more layers of cells, including the amacrine, bipolar, and horizontal cells, will have to be bioprinted in addition to the existing layers, and scaffolds will need to be almost indistinguishable from the human retina.

### 6.3. Conjunctiva

The conjunctiva is a thin, clear, but highly vascularized mucosal tissue that covers the sclera and lines the inside of the eyelids [154,155]. It confers protection and lubrication to the surface of the eye and the palpebra by producing tears and mucus and acts as a barrier to prevent pathogens from entering the eye and infecting its diverse tissues. It is composed of non-keratinized stratified squamous and stratified columnar epithelium. At 3 to 5 cell layers thick, the conjunctiva also contains goblet cells, blood vessels, fibrous tissue, lymphatic vessels, melanocytes, lymphocytes, Langerhans cells, and accessory lacrimal glands dispersed within the epithelial layer [155]. It is followed by a deeper layer of connective tissue, called substantia propria, or conjunctival submucosa, which is unique to the conjunctiva and is formed of fibrous and superficial lymphoid tissue, precisely lymphocytes, mast cells, plasma cells, and neutrophils. Finally, nerves and vessels providing the conjunctiva's innervation and vascularization, respectively, make up the deepest fibrous layer [155].

Conjunctival tissue is susceptible to various inflammatory and autoimmune diseases, as well as injuries, including lacerations, thermal or chemical abrasions, and foreign bodies [10]. The standard treatment for such injuries involves surgery and the transplant of autologous or allograft tissues, such as the amniotic membrane, the innermost layer of the placenta that is in direct contact with the fetus in utero, or pericardium tissue [156–158]. Nonetheless, there are downsides to these usual therapeutic modalities, which encompass the risk of infection, opacification, adverse immune response to the graft, and loss of mucin-secreting goblet cells [159]. Keratinization, a type of metaplasia reaction distinguished by the acquisition of keratin polypeptides and the development of fibrils in the conjunctival epithelium in response to insults or irritation of the ocular surface, is also a possibility [160]. To address these potential concerns, 3D bioprinting of a conjunctiva-like tissue has been proposed as an option, but relatively few studies have been conducted on the matter to date. In 2018, Dehghani et al. used extrusion-based 3D printing to create a membrane suitable for conjunctival regeneration using gelatin, elastin, and hyaluronic acid with human limbal epithelial cells and compared it to conventional amniotic membrane [161]. Following optimization of the bioink with rheological measurements, the printed membrane had suitable color, transparency, and mechanical properties. Analysis of the epithelial cells' biological characteristics, including their in vitro viability, adhesion, and proliferation, was performed and determined to be adequate. The bioprinted gelatin-based membrane was then implanted on injured rabbit conjunctivas for in vitro evaluation. In rabbit models, the epithelialization time for both the amniotic membrane and the gelatin-based printed membrane was comparable, but the results pertaining to inflammation, scar healing, cell density, and granulation tissue formation were superior in the 3D-printed membrane [161].

Despite these positive results, more research is needed to further validate the 3D printing of membranes for conjunctiva regeneration.

Stem cell therapy represents a potential treatment option for ocular surface diseases, and in 2021, Zhong et al. explored this possibility by first expanding rabbit-derived conjunctival stem cells (CjSCs) in vitro and subsequently enclosing these in hydrogel micro-constructs. The stem cell–hydrogel constructs were fabricated using digital light processing (DLP)-based rapid bioprinting to preserve their function and viability [162]. Later, these microscale CjSC-loaded devices were successfully delivered to the epithelium of the bulbar conjunctiva of rabbit eyes via subconjunctival injection, with retained cell viability and the ability to differentiate into goblet cells [162]. The authors advocate for the stem cell delivery approach as a platform for regeneratively treating ocular surface diseases. The following year, the same team used the same bioprinting technique to manufacture a 3D multicellular in vitro model of a pterygium, an overgrowth of vascularized conjunctival tissue that invades the cornea and can negatively affect the vision [163,164]. The microenvironment of pterygium was mimicked with human CjSCs, immune cells, and vascular cells [163]. Both studies showed the potential of 3D-printed stem cells for research in eye surface disorders, and further research will help elaborate translation to therapeutics.

## 7. Limitations of Ocular 3D Printing and Next Steps

### 7.1. Bioprinting Challenges

The application of 3D printing in ophthalmology has grown considerably, with opportunities to create personalized treatment devices and bioprinted tissues for regenerative medicine. However, there remain challenges and limitations to overcome before clinical translation and implementation. First, the achievable resolution with modern 3D bioprinters will still require improvement to reproduce the fine details of ocular structures, such as the retinal layers and microvasculature. Although LIFT bioprinting has been demonstrated to print as small as 10 μm, the diameter of cone cells is less than 4 μm in the fovea, while rods range from 3 to 5.5 μm [165]. In combination with the intricate retinal architecture of more than 130 million cells, accurate reproduction of the human retina remains a future aim, requiring significant advancements in printing technology [128]. Furthermore, more research is needed into the use of multiple cell types in retinal models and the study of cell signaling within the printed retinal scaffold. Future studies should examine in vivo transplantation of bioprinted retinal grafts into animal models as a step towards treating patients. Similarly, although studies have been able to print single layers of the cornea in tissue models, the full multilayer configuration of the cornea remains a challenge to reproduce. Additional investigation is needed into replicating the optical and cellular properties of the multilayered cornea with transplant studies.

Another key barrier to clinical application is the challenge of producing vasculature and innervation in bioprinted tissues to sustain them following transplantation. Efforts are being made to use angiogenic growth factors within bioinks, direct vessel printing, and the embedding of microchannels in printed tissues to enable nutrient diffusion [166]. Although significant advances have been made in tissue models, further fundamental research is required to print entire functional ocular structures such as the cornea or the retina with all their cellular diversity and physiological functions. Future developments in reproducing the structural heterogeneity of organs and providing them with functional vasculature and innervation will bring us closer to clinical translation.

### 7.2. Material Properties

Furthermore, there are areas for improvement in the material properties of 3D printing. First, additional research is required to develop biocompatible materials for ocular devices, and safety studies are needed before clinical implementation. For example, ideal materials for ocular prostheses need to be non-immunogenic with modifiable transparency and flexibility while being mechanically robust. These materials should be evaluated in future studies that assess the long-term incidence of complications and patient quality of life with

their printed prosthesis. In addition, common polymeric materials used in 3D printing are often too hard or brittle to emulate the soft tissue and fluid compartments of the eye. The relatively limited material selection in 3D printing compared to traditional methods means finding the right material to simulate the unique anisotropy of biological tissues is more challenging. To enable accurate training in surgical simulation and dissection models, the development of biomimicking materials with customizable directional strength is needed. In cases such as structural implants where material uniformity is desired, it is also important to consider the anisotropy associated with techniques such as FDM printing. Instead, isotropic printing techniques such as SLA may be selected for these applications to ensure equal directional properties. Regarding bioprinting, continued innovation in bioink preparation is also required to achieve structural integrity while also providing an environment that supports cellular growth. Research is being conducted into the chemical functionalization of hydrogels to match native conditions in the eye and the integration of growth-promoting factors, which can improve cell survival and differentiation [167]. Complementary development of composite scaffolds and stimuli-responsive biomaterials that facilitate tissue remodeling may serve as the foundation for tomorrow's ocular bioprinting advances [150,152,167].

### 7.3. Time and Cost

The variable printing times and substantial start-up costs associated with 3D printing comprise a substantial barrier to its adoption in medicine. First, as printing time increases with the size and complexity of a product, detailed models such as surgical guides can take hours to days to print. This reduces the viability of 3D printing for emergency medical procedures, such as orbital reconstruction after acute injury. The pre-print preparation required from imaging to CAD model creation further contributes to this difficulty. Second, the high capital costs associated with printers and materials act as a barrier to entry for health systems. Investment in medical 3D printing is challenged by low production volumes associated with printing personalized medical devices. This reduces the cost advantages that come from economies of scale and high-volume manufacturing. As a result, clinicians and patients may not be inclined to adopt 3D-printed products over traditional methods without clear clinical benefits and a reduction in costs. These challenges are starting to be addressed with the fall of 3D printing costs as patents expire and the technology matures [166]. Although 3D printing may not compete with manufacturing techniques like injection molding for large-scale applications, it can be highly cost-effective for small-scale personalized products in medicine. In fact, 3D printing often has higher throughput than traditional methods for manufacturing custom small to medium-sized objects, as retooling or creating new molds is not necessary. Furthermore, financial analyses have shown that the reduced operating time enabled by 3D printed models creates notable downstream value to health systems despite initial implementation costs [168]. In ophthalmology, 3D-printed surgical guides can be employed during preparation for non-emergent procedures to improve operative efficiency and patient outcomes [47–49]. Advancements in printing technology, such as DLP and CLIP, have also achieved printing speeds that can meet the needs of the fast-paced medical field [11]. The unique benefits of 3D printing for custom fabrication are well-suited to ophthalmology applications such as ocular prostheses, where it is superior to traditional methods in terms of cost and time. Future innovations such as integrating medical scanners with 3D printers for automated model creation, rapid printing for scalability, and software advancements can continue to increase the accessibility of 3D printing in ophthalmology and medicine.

## 8. Conclusions

In conclusion, 3D printing has opened novel pathways for innovation in ophthalmology. The technology has diverse applications, from the fabrication of patient-specific implants to drug-delivering implants to anatomical models that can aid surgeons and educate trainees. The capabilities of 3D printing for custom manufacture enhance ophthalmic

care as devices can be tailored to the individual variations of patients' ocular anatomy. Moreover, the recreation of intricate ocular structures that is possible with modern techniques has proven highly valuable for training, pre-operative planning, and improving surgical efficiency. Development in the exciting field of bioprinting has also yielded 3D-printed ocular tissues that promise to advance research efforts in regenerative treatments for corneal, retinal, and conjunctival diseases.

The next steps for 3D printing in ophthalmology include overcoming current limitations in achievable complexity, biocompatible materials, low standardization, and cost-effectiveness. As 3D printing technology and research continue to evolve, there will be an acceleration in the solutions available to benefit patients in need. Continued collaboration between clinicians, scientists, engineers, and industry will contribute to overcoming current obstacles and bringing 3D printing closer to routine clinical practice.

The combination of 3D printing and ophthalmology is synergistic, offering impressive opportunities for enhanced patient care, surgical precision, and educational advancements. As we navigate the complexities of this rapidly evolving field, the adoption of 3D printing technologies promises to shape the future of ophthalmic innovation and improve ocular health.

**Author Contributions:** Conceptualization, N.L., M.G. and K.Y.W.; Literature review and data analysis, N.L. and M.G.; Writing—original draft preparation, N.L. and M.G.; Figure visualization, N.L.; Data curation, N.L. and M.G.; Writing—review and editing, N.L., M.G. and K.Y.W. All authors have read and agreed to the published version of the manuscript.

**Funding:** This research received no external funding.

**Data Availability Statement:** No new data were created or analyzed in this study. Data sharing is not applicable to this article.

**Conflicts of Interest:** The authors declare no conflict of interest.

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
