# Peer review of "The Third Dimension of Eye Care: A Comprehensive Review of 3D Printing in Ophthalmology"

_2813-6640, doi:10.3390/hardware2010001_

Round 1
Reviewer 1 Report
Comments and Suggestions for Authors
Very detailed review. It would be interesting to explore more of the limitations on 3D printing such as 1. What is the throughput and how does that impact scalabale manufacturing, 2. are there any directional properties that could have potential impacts and availability of materials for printing.
Author Response
Thank you for your suggestions and review of our manuscript.
We have expanded upon our limitations of ocular 3D printing section to include points on throughput, scalability, and directional properties under our "Time and Cost" and "Material Properties" sections. The changes are highlighted in our resubmitted manuscript for your viewing.
We appreciate your time in appraising our work.
Reviewer 2 Report
Comments and Suggestions for Authors
This paper mentioned three-dimensional (3D) printing in ophthalmology and outlines applications like prosthetic eyes, orbital implants, creating anatomical models, and assisting with surgical planning. Author mentioned that this technology also can apply to new drug delivery platforms for ocular pathologies.
Author Response
Thank you for your review of our manuscript. We appreciate your time in assessing the work and are glad you find it suitable for publication.
Reviewer 3 Report
Comments and Suggestions for Authors
Authors did a very good research work studying the 3D Printing technologies in Ophthalmology. The overall manuscript is well written and clear to the readers.
The novelty of the research done is evident throughout the manuscript.
I have no reccomendations for revisions.
Author Response
Thank you for your review of our manuscript. Though the ratings provided don't seem to match your comments, we appreciate the time you took to assess our work and are glad you find it suitable for publication.
Reviewer 4 Report
Comments and Suggestions for Authors
I understand that this paper summarizes reports on eye care regarding the development trends of medical devices using 3D modeling methods. I have no objection to the content of the paper, but I thought that the caption position in Table 1 should be corrected.
Other comments:
1. What is the main question addressed by the research?
This review summarizes trends in the development of medical devices using 3D modeling methods, and discusses whether 3D modeling methods are effective in medical device development.
2. What parts do you consider original or relevant for the field? What specific gap in the field does the paper address?
This paper is concerned with the fields of biomedical engineering and polymer processing, and is organized from the perspective of polymer processing, which is rarely discussed in biomedical engineering.
3. What does it add to the subject area compared with other published material?
Among medical devices, there are very few reviews that have been compiled with a focus on the eye, and in this respect, they are highly original.
4. What specific improvements should the authors consider regarding the methodology? What further controls should be considered?
I have no objection to the content or organization of the paper. I believe the paper can be published as is, but the position of the caption in Table 1 may need to be corrected.
5. Please describe how the conclusions are or are not consistent with the evidence and arguments presented. Please also indicate if all main questions posed were addressed and by which specific experiments.
The effectiveness and future prospects of the 3D modeling method raised in this paper are summarized in the conclusion, which is not disputed.
6. Are the references appropriate?
References are appropriate.
7. Please include any additional comments on the tables and figures and quality of the data.
Author Response
Thank you for your review of our manuscript. We appreciate your detailed feedback and have made the change to Table 1's caption to appear before the table as suggested.